# Factorized Neural Processes for Neural Processes: $K$-Shot Prediction of Neural Responses

**R. James Cotton,[1,a] Fabian H. Sinz,[2-5,b,*] Andreas S. Tolias,[4-5,c,*]**

[1] Department of PM&R, Northwestern University, Shirley Ryan Ability Lab
[2] Institute for Bioinformatics and Medical Informatics, University Tübingen, Germany
[3] Bernstein Center for Computational Neuroscience, University of Tübingen, Germany
[4] Department for Neuroscience, Baylor College of Medicine, Houston, TX, USA
[5] Center for Neuroscience and Artificial Intelligence, Baylor College of Medicine, Houston, TX, USA

[a]rcotton@sralab.org, [b]fabian.sinz@uni-tuebingen.de, [c]astolias@bcm.edu
[*]equal contribution

## Abstract

In recent years, artificial neural networks have achieved state-of-the-art performance for predicting the responses of neurons in the visual cortex to natural stimuli. However, they require a time consuming parameter optimization process for accurately modeling the tuning function of newly observed neurons, which prohibits many applications including real-time, closed-loop experiments. We overcome this limitation by formulating the problem as $K$-shot prediction to directly infer a neuron's tuning function from a small set of stimulus-response pairs using a Neural Process. This required us to developed a Factorized Neural Process, which embeds the observed set into a latent space partitioned into the receptive field location and the tuning function properties. We show on simulated responses that the predictions and reconstructed receptive fields from the Factorized Neural Process approach ground truth with increasing number of trials. Critically, the latent representation that summarizes the tuning function of a neuron is inferred in a quick, single forward pass through the network. Finally, we validate this approach on real neural data from visual cortex and find that the predictive accuracy is comparable to — and for small $K$ even greater than — optimization based approaches, while being substantially faster. We believe this novel deep learning systems identification framework will facilitate better real-time integration of artificial neural network modeling into neuroscience experiments.

## 1 Introduction

There is a long and rich history of modeling the response of visual cortex neurons to stimuli extending back to the work of Hubel and Wiesel on simple and complex cells [1]. In recent years, artificial neural networks (ANNs) have achieved state-of-the-art performance predicting neural responses to natural stimuli [2–11]. These models are accurate enough that the stimuli that maximally excite a neuron can be computed *in silico*, and when tested *in vivo* indeed drive neurons effectively [11, 12]. However, these approaches place the computational burden of optimizing network parameters *after* extensive data from a neuron has been collected, which prohibits their use in real-time closed-loop experiments. To avoid this optimization step, we wanted a model that can predict the response of a novel neuron to any stimulus, conditioned on a set of $K$ observed stimulus-response pairs – essentially performing $K$-Shot prediction on neural responses.

Garnelo et al. [13] aptly describe how Neural Processes (NPs) can help solve this problem: *"Meta-learning models share the fundamental motivations of NPs as they shift workload from training time to test time. NPs can therefore be described as meta-learning algorithms for few-shot function regression"*. NPs achieve this by embedding input and output measurements into a latent space that maps to a space of functions, essentially learning the distribution over functions and a method to infer the posterior over functions given limited samples [13–15].

A significant advance in modeling visual responses with ANNs was using convolutional neural networks with a factorized readout between the tuning function's location and properties [4, 7]. We found that NPs struggle to learn the space of tuning functions from stimulus-response samples without such a factorized representation. Thus, we developed a Factorized Neural Process (FNP), which is composed of stacking multiple NPs. A key insight for this was that by passing the latent variable computed by early layers to deeper layers, in addition to the observations, we could obtain a factorized latent space while retaining the representational power and efficiency of NPs. We used a two-layer FNP applied to visual responses, where the first NP produces a latent variable for the tuning function's location that the second NP uses to infer the tuning function's properties. We found that a FNP trained on simulated data generalizes to new neurons, successfully inferring the tuning function's location and properties and predicting the responses to unseen stimuli. An FNP trained on neural responses from the mouse primary visual cortex made predictions with comparable accuracy to state-of-the-art approaches, and made these predictions almost *100 times faster*.

In short, our contributions in this work include: ❶ We reformulate the problem of predicting the response of neurons to visual stimuli as a K-shot regression problem, removing the time consuming step of optimizing network parameters for each newly acquired neuron. ❷ We develop a Factorized Neural Process that embeds the observed stimuli-response pairs into a latent space representing the tuning function that is partitioned into location and tuning function properties. ❸ We train this Factorized Neural Process for Neural Processes end-to-end on simulated data and show it approaches the ground truth predictions as the size of the observation set increases. ❹ We found that this approach performs comparably to state-of-the-art predictive models on responses from mouse visual cortex neurons while improving estimation speed by multiple orders of magnitude. The code is available at https://github.com/peabody124/fnp_neurips2020.

## 2 Neural Processes for Neural Processes

The core steps that allow a NP to efficiently meta-learn a $K$-shot regression model are (1) encoding each element from a set of observed input-output observations into a representation space, (2) aggregating the elements in that representation space (typically by taking the mean) to produce a sufficient statistic of the observed set, (3) a conditional decoder that maps the aggregated representation to a function used for prediction of new observations, and (4) training this over many different sets of observations from different sample functions, i.e. meta-learning the distribution over tuning functions [14]. Our approach is largely based on Garnelo et al. [13], which expanded on Garnelo et al. [14] by introducing a stochastic variable used by the conditional decoder. NPs were further extended to include attention in Kim et al. [15], which we do not use in this work.

First, we describe the data generation process we seek to model: Let $\mathcal{F} : \mathcal{X} \to \mathcal{Y}$ be the space of all tuning functions that map from images to neural responses. An individual neuron corresponds to a sample function, $f \in \mathcal{F}$, from which we get $K$ observations $O_K = \{(\boldsymbol{x}_i, y_i)\}_{i=0}^{i<K}$ where $y_i \sim f(\boldsymbol{x}_i)$, $\boldsymbol{x} \in \mathcal{X}$ and $y \in \mathcal{Y}$. The model should maximize the likelihood of a new input-output observation, called the target: $p_\theta(y_t|\boldsymbol{x}_t, O_K)$ with $y_t \sim f(\boldsymbol{x}_t)$, i.e. $K$-shot prediction.

Following the formulation of stochastic NPs [13], this predictions is split into *encoding* the set of observations into a posterior distribution over a latent space, $p_\theta(\boldsymbol{z}|O_K)$ with $\boldsymbol{z} \in \mathbb{R}^D$, and *conditionally decoding* it with a predictive distribution conditioned on the latent variable and target input, $p_\theta(y_t|\boldsymbol{x}_t, \boldsymbol{z})$:

$$p_\theta(y_t|\boldsymbol{x}_t, O_K) = \int p_\theta(y_t|\boldsymbol{x}_t, \boldsymbol{z}) \, p_\theta(\boldsymbol{z}|O_K) \, \mathrm{d}\boldsymbol{z} \qquad (1)$$

The encoder computes the distribution over $\boldsymbol{z}$ using a learned embedding of individual observations that are aggregated into a sufficient statistic: $s_K = \frac{1}{K} \sum_{i<K} h_\theta(\mathbf{x}_i, y_i)$, with $h_\theta : \mathcal{X} \times \mathcal{Y} \to \mathbb{R}^{D_2}$ implemented as a neural network. The dimensionality of this statistic, $D_2$, does not need to match the dimensionality of the latent variable, $D$, but it is used to parameterize a distribution for the

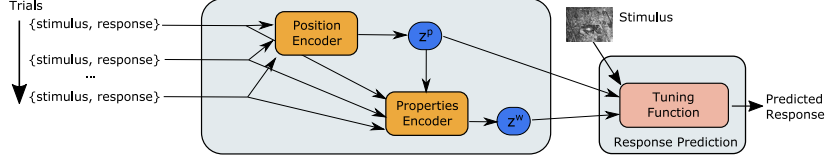

Figure 1: Overview of K-shot neural prediction using a Factorized Neural Process.

latent variable, such as Gaussian: $p_\theta(\boldsymbol{z}|O_K) \sim \mathcal{N}(\mu_\theta(s_K), \sigma_\theta(s_K))$. The decoder function is also implemented as a distribution that is parameterized by the output of a neural network, in our case determining the mean rate of a Poisson distribution $\lambda_\theta : \mathcal{X} \times \mathbb{R}^D \to \mathbb{R}$ and $p_\theta(y_t|\boldsymbol{x}_t, \boldsymbol{z}) \sim$ Poisson$[\lambda_\theta(\boldsymbol{x}_t, \boldsymbol{z})]$.

Our formulation differs from [14, 15] in a few details. First, they used a variational approximation to train the encoder. Because of our approach to efficient K-Shot training, described below in Section 2.7, that involves predicting single samples we can directly optimize Eq. 1 with samples of $\boldsymbol{z}$. Second, and more critically, for NPs applied to image completion, individual observations are pixels with $\mathcal{X} \in \mathbb{R}^2$ and $h_\theta$ was implemented with an MLP [13–15]. To apply this to direct regression of neural responses, $h_\theta$ must embed entire stimulus-response associations where the domain of $\mathcal{X}$ are images $\mathbb{R}^{\text{width} \times \text{height}}$.

We attempted to apply a NP to predicting visual neuron response by including a convolutional neural network (CNN) in $h_\theta$ to extract the relative stimulus features and optimizing Eq. 1. However, we could not achieve good predictive performance despite trying numerous architecture alterations. This motivated us to develop the Factorized Neural Process.

## 2.1 Factorized Neural Process

Prior work using neural networks to model visual responses has demonstrated the importance of identifying the location of a receptive field before trying to model the exact properties. For example, early work used a two stage approach which first computed a spike-triggered average to localize the receptive fields and then stimuli were cropped and centered on this location to build predictive models [6]. In Section 2.4 we make a link between a spike-triggered average and our position encoder. When using CNNs to predict neural responses, Klindt et al. [4] showed it is helpful to factorize the tuning function into a location and the tuning properties within that location. We used a similar approach to factor our latent space into visual location and response properties.

Aggregation in NPs should be invariant to permutation of the order of elements in the set, because changing the order inputs are presented to the function should not change the representation of that function (at least under the generative model of our data stated above). Aggregation using the mean, such as used in the NPs above, is one aggregation operation with this property. Zaheer et al. [16] demonstrated that it applies to a class of deeper network architectures, which they termed Deep Sets. These assure permutation invariance when composing multiple operations, provided each operation (layer) transforms and passes individual elements from a set along with the result of an invariant aggregation operation (such as mean). This insight encouraged us to develop a FNP by composing multiple NPs.

In a FNP, the encoder, $h_\theta^l$, for each layer $l$ receives both observed samples and the latent variable samples from lower layers, $z^l$ (which is an empty set for the first layer):

$$s_K^l := \frac{1}{K} \sum_{i<K} h_\theta^l \left(\boldsymbol{x}_i, y_i, \left\{\boldsymbol{z}_K^j\right\}^{j<l}\right), \qquad \boldsymbol{z}_K^l \sim p_\theta^l\left(\boldsymbol{z}_K^l|s_K^l\right) \tag{2}$$

The decoder receives all the latent variables, which can be marginalized out to give the predictive model for a $L$ layer FNP:

$$p_\theta(y_t|\boldsymbol{x}_t, O_K) = \int_{\boldsymbol{z}} p_\theta\left(y_t|\boldsymbol{x}_t, \left\{\boldsymbol{z}_K^j\right\}\right) \prod_{l=0}^{l<L} p_\theta^l\left(\boldsymbol{z}^l|s_K^l\right) \mathrm{d}\boldsymbol{z}_K^0 ... \mathrm{d}\boldsymbol{z}_K^{L-1} \tag{3}$$

## 2.2 FNP Model for Visual Responses

Following the motivation above, we use a two-layer FNP to partition the latent space into the tuning function position, $z_K^p$, and the tuning function properties within that location, $z_K^w$ (Figure 1). The factorized probability (following Eq. 3) is thus:

$$p_\theta(y_t|\boldsymbol{x}_t, O_K) = \int_{z_K^p, z_K^w} p_\theta\left(y_t|\boldsymbol{x}_t, z_K^w, z_K^p\right) p_\theta\left(z_K^w|s_K^w\right) p_\theta\left(z_K^p|s_K^p\right) \mathrm{d}z_K^p \mathrm{d}z_K^w \qquad (4)$$

The latent space acquires this factorization from the inductive bias in the architecture of the conditional decoder (Section 2.6), which uses a similar factorization between tuning position and properties as Klindt et al. [4]. With this conditional decoder, we empirically found certain architectures helped the encoders learn an appropriate embedding for these latent variables, which we describe in Sections 2.4 and 2.5. All components use a common Group Convolutional Neural Network to compute the relevant features space from the stimuli, which we will describe next.

## 2.3 Group Convolutional Network

One of the most striking features of V1 neurons are their sensitivity to stimulus orientation [1], so CNN-based models expend a great deal of capacity representing the same feature at different orientations. Cohen and Welling [17, 18] developed Group Convolutional Neural Networks (G-CNNs), which are rotationally equivariant network that reuse parameters across orientations, similarly to how CNNs reuse parameters across space. G-CNNs have improved performance on a number of image classifications tasks [19–21] including the modeling of visual cortex neurons [22, 23]. They have also been combined with a DenseNet [24] architecture when modeling histology images [25]. Our stimuli were grayscale images, $\mathcal{X} \in \mathbb{R}^{H_x \times W_x}$ that were transformed to a reduced spatial dimension with multiple feature maps: $g_\theta : \mathbb{R}^{H_x \times W_x} \to \mathbb{R}^{H \times W \times C}$.

Our G-CNN Dense Blocks followed the bottleneck DenseBlock architecture of [24] (other than using group convolutions) of 1) Batch Normalization [26], 2) non-linearity, 3) $1 \times 1$ group convolution, 3) Batch Normalization, 4) non-linearity, 5) spatial group convolution, 6) concatenating this output to the inputs. Specific architectural parameters are described in the experiments. Following Bekkers et al. [20] and Veeling et al. [25], images were passed through an initial layer that lifted them into a group representation, followed by the G-CNN Dense Blocks, and a subsequent projection from the group representation (i.e. a tensor with an additional dimension representing orientation) down to a representation over image space by flattening the group and channel dimensions together. An additional $1 \times 1$ convolution [27, 28] reduce this to the final channel depth. We found using the G-CNN DenseNet architecture resulted in faster and more reliable training than a standard CNN, although we did not perform exhaustive searches over architectures.

## 2.4 Tuning function position encoder

The first layer is designed to produce a distribution for a latent variable centered at the location of the tuning function, $z_K^p$. The latent has two dimensions corresponding to horizontal and vertical position. To compute this, we found it was important to preserve spatial structure from $g_\theta(\boldsymbol{x})$ when performing the set aggregation, which we did by averaging the embedded observations into an image analogous to a trainable, non-linear, spike-triggered average:

$$s_K^p = \frac{1}{K} \sum_{i < K} h_\theta^p\left([g_\theta(\boldsymbol{x}_i), y_i g_\theta(\boldsymbol{x}_i), y_i]\right), \qquad s_K^p \in \mathbb{R}^{H \times W}$$

Where $[\cdot, \cdot]$ concatenates on the channel dimension (and tiles non-spatial values, such as $y_i$, along the spatial dimension) and $h_\theta^p$ is a series of $1 \times 1$ convolutions that outputs a single channel; the first two $1 \times 1$ retain the same channel depth as $g_\theta$ followed by an ELU activation [29]. We interpret softmax($s_K^p$) as a spatial distribution over the receptive field location of a given neuron. In order to sample a single spatial readout point, we separately computed the marginal means $\boldsymbol{\mu}(s_K^p)$ and standard deviations $\boldsymbol{\sigma}(s_K^p)$ of softmax($s_K^p$), and used them to parameterize a truncated normal distribution $\mathcal{N}_|$, cut at the same height and width as the output of the G-CNN ($H \times W$):

$$z_K^p \sim \mathcal{N}_|\left(\boldsymbol{\mu}(s_K^p), \boldsymbol{\sigma}(s_K^p)\right), \qquad z_K^p, \boldsymbol{\mu}(s_K^p), \boldsymbol{\sigma}(s_K^p) \in \mathbb{R}^2.$$

We also experimented with learning an MLP to transform $s_K^p \to (\boldsymbol{\mu}, \boldsymbol{\sigma})$. While this worked, it was much slower to train than the Gaussian approximation of softmax($s_K^p$) we used.

## 2.5   Tuning function properties encoder

The second layer of the FNP computes the tuning function's properties at the location determined by the first layer:

$$s_K^w = \frac{1}{K} \sum_{i<K} h_\theta^w \left( [\mathcal{T} \left( g_\theta(\boldsymbol{x}_i), \boldsymbol{z}_K^p \right), y_i] \right), \tag{5}$$

where $\mathcal{T} : \mathbb{R}^{H \times W \times C} \times \mathbb{R}^2 \to \mathbb{R}^C$ is a differentiable operator that performs interpolation on the image features at a particular location to preserve differentiability with respect to the spatial location. We use a spatial transformer layer for that [30]. Essentially, (5) selects the features computed by the G-CNN at a given location, concatenates them with the response, and passes them through a two-layer MLP, $h_\theta^w$.

We split $s_K^w = (\boldsymbol{\mu}_K^w, \boldsymbol{\Sigma}_K^w)$ into a mean and covariance (more specifically the vectorized lower triangular component of the covariance) and use it to parameterize the distribution over the $D$-dimension latent variable for tuning function's properties within that location:

$$\boldsymbol{z}_K^w \sim \mathcal{N}(\boldsymbol{\mu}_K^w, \boldsymbol{\Sigma}_K^w), \qquad \boldsymbol{z}_K^w, \boldsymbol{\mu}_K \in \mathbb{R}^D.$$

## 2.6   Conditional Decoder

The latent variables for tuning function position and tuning function properties, $\boldsymbol{z}_K^p$, $\boldsymbol{z}_K^w$, summarize the observed set of stimulus-response pairs, $O_K$. They define a $K$-shot regression tuning function that we use them to predict the response to new target stimuli, $\boldsymbol{x}_t$.

$$\lambda(\boldsymbol{x}_t, \boldsymbol{z}_K^p, \boldsymbol{z}_K^w) := \texttt{ELU} \left( [\mathcal{T} \left( g_\theta \left( \boldsymbol{x}_t \right), \boldsymbol{z}_K^p \right), 1] \cdot u_\theta \left( \boldsymbol{z}_K^w \right) \right) + 1$$

Similar to prior work [4, 5, 7, 11, 22], our response predictor uses a linear weighting of the features from the convolutional network selected at the neuron's tuning function's location. Because we sometimes used a different dimensional representation for tuning function's properties than the feature map, we used an additional two-layer MLP, $u_\theta$ to match the dimensions. We also found this additional transformation of the latent improved performance, although did not experiment with this extensively. The predictive response probabilities for $y_t$ are modeled as a Poisson distribution with mean rate $\lambda$:

$$p_\theta(y_t|\boldsymbol{x}_t, \boldsymbol{z}_K^p, \boldsymbol{z}_K^w) \sim \texttt{Poisson}[\lambda(\boldsymbol{x}_t, \boldsymbol{z}_K^p, \boldsymbol{z}_K^w)] \tag{6}$$

## 2.7   Efficient K-Shot Training

Each training sample for this model consists of a set of K stimuli and responses with an additional target pair and it must be trained over many sets. If implemented naïvely and sampling a range of set sizes, training is computationally prohibitive as it requires passing thousands of images through the G-CNN per sample. Additionally, in this case the loss is determined by the prediction of a single sample, thus the gradient is fairly noisy. To greatly accelerate training we used two techniques.

The first technique was including the responses of multiple neurons to the same stimuli in a minibatch. This allowed the computation of $g_\theta(\boldsymbol{x})$ to be reused across the neurons, at the expense of GPU memory. This memory is proportional to the number of stimuli $\times$ the number of neurons, because Eq. 5 is computed in parallel for all neurons and must be stored during each step to allow backpropagation through the G-CNN.

The second technique was for a set of $T$ trials, we computed the $K$-shot prediction for every set size up to $T-1$ by predicting the response on the next trial. Because aggregation involved computing the mean outputs along the set dimension, this could be efficiently implemented with a cumulative sum that excluded the current element divided by the set size. This allowed us to test $T-1$ predictions rather than 1 for each training set with almost no increase in computation. Thus the per-neuron loss function we minimized was:

$$\mathcal{L}(\theta) = - \sum_{k=0}^{T-1} \log p(y_{k+1}|\boldsymbol{x}_{k+1}, O_k)$$

With the likelihood approximated by the FNP described in Eq. 4. We do not compute the full marginalization in Eq. 4 but estimate it using Monte Carlo samples from $\boldsymbol{z}_k^w$ and $\boldsymbol{z}_k^p$ drawn for each value of $k$. The distribution parameters are optimized using the reparameterization trick [31].

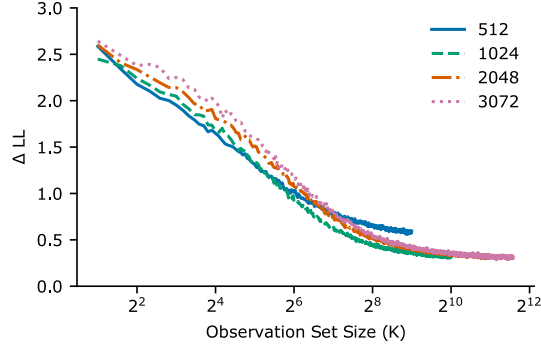

Figure 2: Predictive accuracy versus ground truth. Legend indicates the maximum observation set used during training. See Predictive Accuracy versus observation set size for the definition of $\Delta LL$.

In prior work with NPs (that had much less computationally expensive inputs), the full $T$ trials were randomly partitioned into the observation and target set to learn generalization over set size [13–15]. We leave it to future work to determine if this can be efficiently done with our model, for example generating multiple splits for the encoded values in a single batch, and if it improves training speed or final performance.

## 3 Experiments with Simulated Neural Responses

We first validated that using our approach, a trained FNP could infer the ground truth tuning function from simulated simple and complex visual neurons. Experiments using different architectures and hyperparameters were managed using DataJoint [32]. Please see the Appendix for the architecture details used in the presented results.

### 3.1 Simulated responses

Simulated neurons had a linear component to their receptive field that was generated using a Gabor kernel, $k_\phi$, with parameters $\phi$ including $x$ and $y$ location, orientation, frequency, width, phase offset and scale. Cells could either be simple or complex. Responses for simple cells were generated with the inner product between the receptive field kernel and the stimuli, followed by a ReLU non-linearity. Complex cells used an energy-based model which involved a second kernel, $k_{\phi+}$, with the same parameters as the first but with the phase offset by $\pi/2$. Responses were then sampled from a Poisson distribution for a given cell.

$$\lambda_\phi(\boldsymbol{x}) = \begin{cases} \text{ReLU}\left(\boldsymbol{x} \cdot k_\phi\right) & \text{, if } \phi \text{ simple} \\ \sqrt{\left(\boldsymbol{x} \cdot k_\phi\right)^2 + \left(\boldsymbol{x} \cdot k_{\phi+}\right)^2} & \text{, if } \phi \text{ complex.} \end{cases}, \qquad r_i \sim \text{Poisson}[\lambda_\phi(\boldsymbol{x}_i)]$$

To simulate the finite data available in real experiments, we used a fixed set of 5000 neuron parameters and 10,000 stimuli in the training data and a different set in the validation data. Stimuli were sampled from ImageNet downsampled to $16 \times 16$ or $32 \times 32$ [33, 34]. Each sample used a random stimulus order with a fixed number of trials.

### 3.2 Predictive Accuracy versus observation set size

To quantify the predictive performance of this model for different numbers of observations, we computed the difference between the negative log-likelihood of the response under the $K$-shot predictive distribution to the ground truth model. This was averaged across many neurons:

$$\Delta LL_k = \langle -\log p_\theta\left(y_{k+1}|\boldsymbol{x}_{k+1}, O_k\right) + \log p\left(y_{k+1}|\lambda_\phi\left(\boldsymbol{x}_{k+1}\right)\right) \rangle$$

We found that as as the size of the observation set increased, the predictive accuracy improved up to several hundred observations and then began to saturate. We also found that increasing the maximum set size used during training had a slight benefit in the asymptotic performance when increasing

from 512 to 1024 trials, but with little benefit beyond this (Fig. 2). These were averaged over three different seeds, with each fit producing similar performance.

Despite trying a number of architecture variations, we could not get the asymptotic performance to quite reach ground truth. However, it performed well, with a $\Delta LL$ of $0.4$ corresponding to a correlation coefficient between the ground truth mean response, $\lambda_\phi(\boldsymbol{x})$, and the model prediction mean, $\lambda_{K=1024}(\boldsymbol{x})$, of $0.8$.

### 3.3 Latent variables accurately capture the tuning function

We confirmed the information about the tuning function was correctly factorized by computing the correlation coefficient between the latent variable for location, $\boldsymbol{z}_k^p$, and the ground truth location of the kernel, $k_\phi$. We found that with only 64 observations there was a correlation of $0.8$, and it reached nearly $1.0$ with 256 observations (Fig. 3a).

We then asked if the latent variables $(\boldsymbol{z}_K^p, \boldsymbol{z}_K^w)$ from 1024 observations were sufficient to reconstruct the receptive field. First, we computed receptive fields as the gradient of the tuning function conditioned on the latent variables:

$$RF_\nabla^K = \nabla_{\boldsymbol{x}}\left([\mathcal{T}\left(g_\theta(\boldsymbol{x}), \boldsymbol{z}_K^p\right), 1] \cdot u_\theta(\boldsymbol{z}_K^w)\right)$$

For simple cells the gradient showed a good correspondence to the kernel used to generate the responses, $k_\phi$ (Fig. 3b). For complex cells it was in the correct location, but did not show the same structure as the kernel. This is expected as complex cells are not well described by a single kernel. We then computed the maximally exciting images (MEIs) for a neuron similarly to Walker et al. [11] by maximizing the predicted response, conditioned on the latent variables sampled after an increasing number of observations:

$$MEI_K = \arg\max_{\boldsymbol{x}}\left([\mathcal{T}\left(g_\theta(\boldsymbol{x}), \boldsymbol{z}_K^p\right), 1] \cdot u_\theta(\boldsymbol{z}_K^w)\right) - \kappa\|\boldsymbol{x}\|$$

With $\kappa = 0.01$ to regularize the images. As desired, MEIs computed with more observations converged towards the ground truth kernels, with complex cells having an anticipated random phase offset.

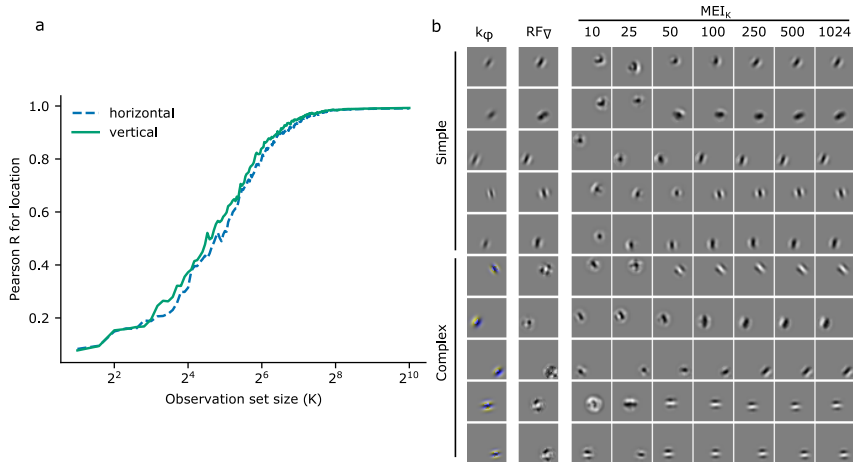

Figure 3: a) the correlation between the location latent variable, $z_k^p$, and the ground truth for increasing observations. b) Reconstruction of receptive fields (RF). Each row corresponds to a different cell with the bottom half being complex cells. The first column shows the ground truth kernels, the second column is the RF reconstructed by the gradient method, and the remaining block shows the maximally exciting images computed using increasing numbers of observations. Ground truth kernels of complex cells use pseudocolor to reflect the two phases in the energy model and any reconstruction of this energy model with the same orientation and location is equally valid, regardless of the phase.

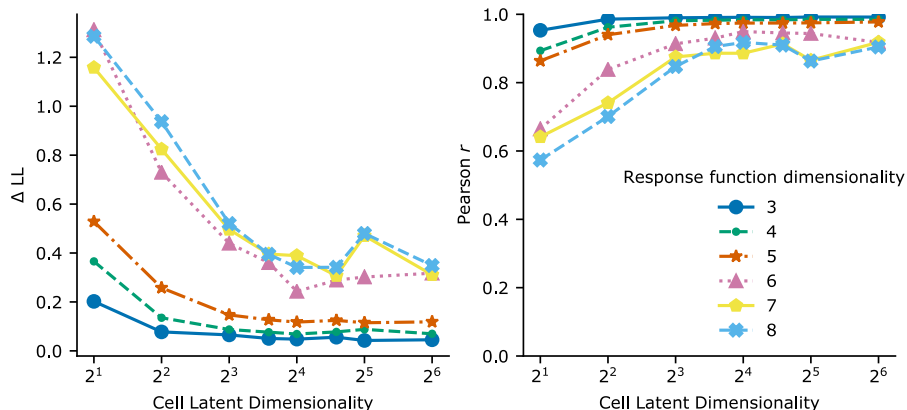

Figure 4: **Left**: The difference between the predictive log likelihood and ground truth as the dimension of the tuning function properties latent variable increases. Each line reflects an increasingly complex RF space. **Right**: The correlation between the ground truth mean firing rate and the predicted mean firing rate for the same data.

### 3.4 Latent dimensionality and stimulus complexity

We also studied how important the tuning function property's latent dimension $D$, with $\boldsymbol{z}_K^w \in \mathbb{R}^D$, was to the predictive performance by increasing it from 2 to 64 (all experiments above used 64). We did this with different complexities of the simulated receptive fields by reducing the number of parameters in $\phi$ that were randomized. In all experiments the orientation and location of the tuning function was randomized ($\phi \in \mathbb{R}^3$). We increased the tuning function dimensions by then additionally randomizing (in order): frequency, width, phase offset, simple only versus simple and complex cells, and scale. Because this analysis involved refitting many models, we performed it with $16 \times 16$ stimuli. We found the performance improved with greater model capacity (tuning function properties latent dimension) and this impact was much more pronounced for more complex (higher dimensional) tuning functions (Fig. 4). Randomizing the phase offset produced the greatest reduction in predictive accuracy, although performance still remained quite good with high correlations between the model predictions and ground truth. Encouragingly, including complex cells did not produce a significant change in performance.

## 4 Experiments with real neural responses

We next tested our approach on real visual responses recorded with the same experimental paradigm as in Walker et al. [11], and found it had a comparable predictive performance to optimization-based approaches. The data consists of pairs of neural population responses and grayscale visual stimuli sampled and cropped from ImageNet, isotropically downsampled to $64 \times 36$ px, with a resolution of $0.53$ ppd (pixels per degree of visual angle). The neural responses were recorded from layer L2/3 of the primary visual cortex (area V1) of the mouse, using a wide field two photon microscope. A single scan contained the responses of approximately 5000–9000 neurons to up to 6000 images.

We trained an FNP on 57,533 mouse V1 neurons collected across 19 different scans and tested it on 1000 neurons from a hold-out scan (i.e. never seen during training). During testing, we assigned the latent variables assigned to their mean values: $\boldsymbol{z}_K^p := \boldsymbol{\mu}(s_K^p)$ and $\boldsymbol{z}_K^w := \boldsymbol{\mu}_K^w$, and used these in Eq. 6. We measured the $K$-shot predictive accuracy for each neuron as the correlation between the predicted mean from the conditional decoder, $\lambda(\boldsymbol{x}_t, \boldsymbol{z}_K^p, \boldsymbol{z}_K^w)$, and the real responses, $y_t$, for the remaining trials. In agreement with synthetic data, the predictive accuracy improves rapidly with the first several hundred trials and continues to improve with additional observations (Fig. 5). We compared the performance of our FNP to an optimization based approach similar to Klindt et al. [4], adapted for mouse V1, which we reference as Per Neuron Optimization (PNO). We measured the predictive performance of PNO similarly to FNP, on the same 1000 neurons with the readout optimized with $K$

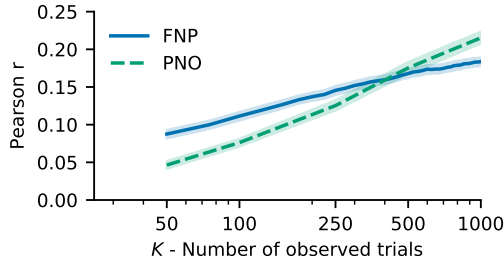

Figure 5: Performance of a FNP for $K$-shot prediction for new neurons compared a traditional approach with per-neuron optimization (PNO) for $K$ up to 1000 trials

trials and used to predict the response to the remaining stimuli. Excitingly, FNP performs well and with 1k images is almost as accurate as PNO (which is optimized for those individual cells), and even *outperforms* it for smaller numbers of observations (Fig. 5). This likely arises because the FNP learns the prior distribution over tuning functions, which has a greater influence with less data. Please see the Appendix for details of both FNP and PNO fitting and testing.

These experiments also demonstrated the speed improvements for inferring the tuning function of a newly recorded neurons that FNP was designed for. While fitting the FNP to the training data took 6 days using two V100 GPUs, computing the latent variables for one thousand neurons with $K = 1000$ took only 250 ms on a 1080Ti. This is in comparison to PNO which takes from 20 s to compute the readout using a pretrained CNN (Supplementary Table 6.3). Thus an FNP is two orders of magnitude faster, enabling real-time inference of tuning functions within the time of a single stimulus presentation.

## 5    Discussion

Using a Factorized Neural Process, we are able to learn a distribution over tuning functions that can be conditioned on a set of observed stimulus-response pairs and predict the response to novel stimuli. We first focused on simulated data from simple and complex cells where we could compare the inferred tuning functions to the ground truth. Importantly, the model performed equally well when including complex cells, which is not possible for classical techniques like spike-triggered average that similarly accumulate sufficient statistics. The fact that the asymptotic log likelihood for predictions did not reach the ground truth also indicates there is room to increase the model capacity, although the correlation between the ground truth and model predictions exceeded $0.8$. Following prior work [4, 7, 11, 22], we restricted ourselves to a decoder that was a factorized linear readout on output of $g_\theta$, but learning a more powerful decoder could also improve the capacity. We then tested our approach on data from the mouse primary visual cortex in response to natural images. We found the trained FNP predicted the responses to test data with comparable accuracy as a model specifically optimized for those neurons, and even exceeded the performance when conditioned on less than 500 trials. Additionally, the FNP made these predictions orders of magnitudes more quickly than an optimization-based approach, thus opening the door to real-time, closed-loop inference of tuning functions updated after every stimulus presentation.

This work was motivated by real-time experimentation, but during an experiment the best way to know how a neuron responds to a stimulus is to measure it. The real need is using the observations to rapidly generate stimuli to test a hypothesis. We envision combining a FNP for rapid inference with a generator network that takes the latent representations as input and is trained *in silico* prior to experiments to generate stimuli to illicit a maximal responses or reduce the uncertainty in the latent representations. We believe this general approach of training a more powerful model prior to experiments that is capable of rapid, real-time inference will be a powerful tool for the neuroscience community, and that this approach using FNPs will facilitate it.

## Broader Impact

We hope this approach will be useful to the Neuroscience community and that Factorized Neural Processes may have even broader applications for modeling functions. The ability to perform real-time, closed-loop experiments and to performances inferences with less data may reduce the amount of time to record from animals or the number of experimental sessions. We do not believe this methodology or the demonstrated application will disadvantage anyone.

## Acknowledgments and Disclosure of Funding

RJC thanks the Research Accelerator Program of the Shirley Ryan AbilityLab for support during residency. FHS is supported by the Carl-Zeiss-Stiftung and acknowledges the support of the DFG Cluster of Excellence "Machine Learning – New Perspectives for Science", EXC 2064/1, project number 390727645. Supported by the Intelligence Advanced Research Projects Activity (IARPA) via Department of Interior/Interior Business Center (DoI/IBC) contract number D16PC00003. The U.S. Government is authorized to reproduce and distribute reprints for Governmental purposes notwithstanding any copyright annotation thereon. Disclaimer: The views and conclusions contained herein are those of the authors and should not be interpreted as necessarily representing the official policies or endorsements, either expressed or implied, of IARPA, DoI/IBC, or the U.S. Government.

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
