[Supplementary Material]

| Parameter ($\phi_i$) | Distribution | Constant |
|---|---|---|
| X and Y location | Uniform spanning inner 2/3 of range | NA |
| Orientation | Uniform$[0, \pi]$ | NA |
| Frequency | $0.08 + 0.16/(1 + e^{\mathcal{N}(0,1)})$ | 0.16 |
| Width | $2 + 1/(1 + e^{\mathcal{N}(0,1)})$ | 2 |
| Phase | Uniform$[0, \pi]$ | 0 |
| Complex? | Bernoulli$[0.5]$ | 0 |
| Scale | $1.0 + 0.5 \cdot \tanh(\mathcal{N}(0,1))$ | 1 |

Table 1: Parameters used for simulated receptive fields

# 6 Appendix

## 6.1 Simulated Responses Tests

### 6.1.1 Architecture

Our G-CNN DenseNet [24] architecture had a lifting convolution layer into the group space with 32 channels and spatial kernel size $11 \times 11$, followed by three Dense Blocks each with a growth factor of $k = 8$ with the $1 \times 1$ convolution having $4k$ channels and kernel sizes $5 \times 5$, followed by two $3 \times 3$ layers. Group convolutions were sampled at 8 equally spaced orientations computed with bilinear interpolation and a circular mask [20, 21]. The group space had a final batch normalization [26] and ELU activation [29] and then was projected into image space by flattening the orientation dimension into the feature channels. A final $1 \times 1$ convolution reduced the features to 64 channels, which we refer to as $C$. Our G-CNN implementation was based on the implementation described in Bekkers et al. [20] found at `https://github.com/tueimage/SE2CNN/tree/master/se2cnn`.

The observation embedding for the location prediction, $h_\theta^p$, was a $1 \times 1$ convolution with three layers, the first two with the same depth as the G-CNN output and the final layer with only one channel. The observation embedding for the tuning function properties latent variable, $h_\theta^w$ and $u_\theta$ were both two layer MLPs networks each with with $C + 1$ channels and no activation function after the final layer of $u_\theta$. The activation function throughout was ELU.

### 6.1.2 Training and Testing

The simulated training data consisted of 5000 neuron parameters used to generate receptive fields and responses as described in Section 3.1. The linear component of the receptive field was generated with scikit-image `gabor_kernel` [35]. The parameters were selected randomly or fixed for the experiments described in Section 3.4. The set of values used for both the training and testing data are shown in Table 1. Stimuli were selected from a fixed 10k images from the training split of ImageNet. The testing performance was measured using images from the testing split of ImageNet using unique tuning function parameter samples.

The experiments in Section 3.3 used image sizes of $32 \times 32$. For the experiments in Sections 3.2 and 3.4 the stimuli were downsampled to $16 \times 16$ due to the number of models trained.

The model was implemented in TensorFlow 2.0 [36] with TensorFlow Probability used for all distributions. We used the ADAM optimizer [37] with learning rate $10^{-3}$ with L2 weight decay $10^{-6}$ based on prior hyperparameter optimization. We used early stopping when the training loss did not improve for 75 steps. Models were fit on a Titan RTX, although for larger stimulus size and models we would distribute across multiple GPUs. To support multiple-GPU training, we created mini-batches of (number of GPU x trials x neurons per GPU) size. This was because TensorFlow treats the first dimension as the batch dimension and splits along that when distributing across GPUs, and so if the trial dimension was the first one the data had a shorter maximum length. Depending on the complexity of the tuning functions and model architecture, fitting took from several hours to several days. Experiments including different architectures and hyperparameters were managed using DataJoint [32].

## 6.2 V1 Responses FNP Architecture and training details

The architecture used for fitting the *in vivo* data was the same as just described for the simulated responses, with the following changes. The G-CNN had a lifting convolution layer into the group space with 16 channels and spatial size of $17 \times 17$, followed by three G-CNN Dense Blocks: the first with spatial dimension $5 \times 5$ with 8 channels, followed by two of size $3 \times 3$ with four channels (resulting in a total of 32 group channels after these four layers). The final $1 \times 1$ convolution reduced to 16 channels. The latent variable had dimension of 24. These parameters were changed primarily to reduce the memory during training while maintaining enough capacity

| Network | Time |
|---------|------|
| FNP Optimization (once) | 6 days on dual V100 |
| FNP Inference for 1000 trials | **250 ms** (1080Ti) |
| PNO Readout only | $\sim 20$ s (1080) |
| PNO CNN+Readout | $\sim 5$ min (1080) |
| PNO + Hyperparameters | $\sim 12$ h (1080) |

Table 2: Comparison of training and inference times for FNP and PNO

for good predictions. Because these experiments used images of $36 \times 64$ they used significantly more memory than the $16 \times 16$ and $32 \times 32$ images used in simulations. Similarly the batch size was limited to only several neurons because each "sample" in a batch is the entire set of $K$ stimuli and responses. The FNP was trained for six days on two V100 GPUs using a batch size of 4 neurons per GPU and a total size of 1024 stimuli.

Neurons and trials were randomly selected from the 57,533 V1 neurons. Each batch used neurons from a randomly selected one of the 19 training scans, due to the need to have responses to the same stimuli. Because the aggregation operation is order invariant, permuting the order *within* the observation batch does not change the inferred latent distribution. However, the ordering was randomized to place individual trials at random $K$ positions for the $K$-shot prediction. The optimizer, learning rate and weight decay were unchanged from our experiments with simulated data.

We also note that the Poisson likelihood is strictly not correct, because the denconvolved fluorescence traces are real while the Poisson distribution is defined on integers. However, following previous work we still used it because it seems to yield better performance in practice [4, 5, 7, 11, 22].

## 6.3 PNO model fitting and testing

We trained it on the responses of a single scan with 4,335 neurons to 5k natural scenes. We used a 4-layer convolutional core with 64 channels in each layers. In the first layer, we used a full convolution with kernel size $15 \times 15$. In the subsequent layers, we use depth separable convolutions with kernel size $13 \times 13$ [38]. We used a ELU nonlinearity [29] and batch normalization before them [26]. We used the same readout as Klindt et al. [4] and as the FNthat factorizes the linear mapping from the final tensor to the neural response into a spatial and a channel component. The final linear mapping was followed by an ELU nonlinarity and an offset of one, to keep the responses consistent. We used L2 regularization on the Laplace filtered input kernels (regularization coefficient $\gamma = 1$), and L1 regularization on the readout coefficients (regularization coefficient $\gamma = 0.0024$). The network was trained using Poisson loss and the ADAM optimizer [37] with an initial learning rate of $0.00414$. We used early stopping based on validation correlation between the model prediction and single trial neural responses using a patience of five (maximum number of steps the validation score is allowed to not improve). Every time, the patience limit was hit, we decreased the learning rate by a factor of 0.3. We repeated this for a total of four times. At the end, we restored the best model over the training run. We trained the full network on the full dataset. Subsequently, we froze the core CNN and fit the readout to 1,000 new neurons on up to 1,000 images and measure the prediction accuracy. We used the same training protocol as above.