[Reviews · NeurIPS 2020]

Review 1

Summary and Contributions: Presents a more efficient method for estimating receptive fields from neurophysiological recording data based on the idea of using a Neural Process to model the probability distribution p(y|x), where y is the neuron's response and x is the input. The model is evaluated on a set of model complex cells and demonstrates the ability to quickly converge on the correct receptive field with many fewer stimulus-response trials.

Strengths: Leverages a sophisticated, state of the art approach from neural networks/machine learning to improve the ability to estimate stimulus-response models for neurons in the brain.

Weaknesses: The model is demonstrated only on model data for a relatively simple type of neuron. But this is a good start to demonstrate the viability of the approach.

Correctness: as far as I can tell

Clarity: yes, maybe could use a gentler introduction to NP theory for those not already familiar.

Relation to Prior Work: yes

Reproducibility: Yes

Additional Feedback:


Review 2

Summary and Contributions: The authors propose a method to infer the stimulus response tunings of neurons given data from few (k-shot) recordings. The goal is to improve the quality of in vivo recording sessions by allowing experimenters to quickly adjust stimuli to suit the neurons that are being recorded. The method leverages the problem's structure in the algorithm design, which uses a factorized Neural Process method. August update: Thank you for a clear, interesting rebuttal. It answered my questions (and questions which I had not thought of, from other reviewers). The info re run-times, and higher correlations at low K, were especially valuable.

Strengths: This is a very interesting paper and set of experiments. The authors cleverly leverage the problem structures' amenability to factorization and G-CNNs. The use-case (tuning stimuli in real-time to increase the quality of in vivo neural recordings) is compelling. The use-case has significant broader impact (ironically not remarked on by the authors): All of neuroscience necessarily rests on a foundation of suffering by experimental animals. Our humane obligation is to inflict the least harm possible, and also conversely to extract the greatest good possible from necessary harm inflicted. Methods that improve the quality of data from a destructive experiment advance this goal.

Weaknesses: The paper is very unclear to me in places. If this were a journal setting, I would accept the paper pending some rewrites for clarity.

Correctness: Yes.

Clarity: The paper is very unclear to me in places. I note that I am not well versed in some of the particular methods. If other reviewers find the exposition clear, I can defer to them.

Relation to Prior Work: Yes, it is well-done.

Reproducibility: Yes

Additional Feedback: My assessment of "accept" assumes that you can carefully assess the paper for clarity (eg 3.3, 3.4) and edit as needed before final submission. Typos: lines 83, 91, 162, 174, fig 3 a) caption. Please number all equations. line 76 and 92: what is q? (I'm guessing P(z | s_k) 108: In what ways is the proposed method more desirable than a 2-stage method as in ref[6]? Is there benefit to an all-in-one method? Eqn (1): does the integration over dz^p give better results than using a simple max likelihood estimate (MLE)? Especially given that z^p is modeled as a gaussian, for which MLE is easy (just the mean). I ask out of general interest, but also due to runtime considerations (see below). One selling point of NPs is that they give measures of uncertainty. I do not see this being called upon by the use-case or proposed method (eg for post-analysis). a) is this used in a way I did not catch? b) are there other reasons to use NPs besides uncertainty measures? Section 3.3: This section lost me. a) what are the dimensions of the various elements: is R^(HxWxC) a vector or an array, and if a vector, how is spatial information preserved. b) what is 'the response' y_t: a time-series, a voltage, a spike (categorical), a spike train, a firing rate? Same issue at line 163. c) line 141: unclear to me. Section 3.4: What are the cell types? Direction oriented, point (donut) activated, tuned to frequency? (info from lines 180 - 185 would be very useful here). I'm guessing "cell types" is not a categorical (though "types" suggests this). Do you mean cell characteristics, with continuous parameters, within a fixed cell type? 152: Are the sampled coordinates not pixel locations? Why do you interpolate to sub-pixel locations? Fig 2: how does the x-axis relate to real experimental constraints (ie what is a real-world k)? How does the y-axis (LL) relate to accuracy of the cell response? 215: fitting runtime = several hours to days: Is this useful for the key use-case of tuning stimuli for in vivo experiments? In some cases a mouse, for example, may be in an electrode (or other) rig for multiple days; but in many cases the neurons being sampled do not deliver data for more than a few hours, or they degenerate considerably over time due to experimental stresses, so faster runtime is important. fig 3 b) Why are the estimated complex cell RFs inverses of the true RFs. Is this due to the offset mentioned in 244? 282: Is code available but the repo is withheld during review? Or is the fact of whether code will be available being withheld? Giving access to a codebase at publication would be a strong asset, but it is not clear whether the authors intend this. Broader impacts: I believe you miss a key point: All of neuroscience necessarily rests on a foundation of suffering by experimental animals. Our humane obligation is to inflict the least harm possible, and also conversely to extract the greatest good possible from necessary harm inflicted. Methods that improve the quality of data from a destructive experiment advance this goal.


Review 3

Summary and Contributions: This paper introduces a method, the Factored Neural Process (FNP), for fitting stimulus-response pairs (e.g. action potentials or firing rates in response to shown image patches) to their true tuning functions. In contrast to existing methods, e.g. learning linear mappings from deep neural network features to neural responses on a training set of images, FNP is designed to be used "online," such that an estimate of a neuron's tuning function can be obtained from a relatively small number of image/response/trial observations. When applied to synthetic neural data modeled after responses common in primary visual cortex, FNP accurately recovers the true tuning functions.

Strengths: This approach is well motivated and would address an important problem, the inference of neural tuning online -- i.e. during a recording session of real neurons, where expensive optimization could limit the types of closed-loop experiments that are possible. The factoring of visual neural tuning into a spatial component and a feature component has proven very effective in predicting neural responses from the learned features of deep neural networks, so the application of this approach to neural processes is a logical addition.

Weaknesses: The main weakness of this work is that the results don't show the method is or will be useful for the thing it was designed for: inferring neural tuning functions more efficiently than existing alternatives, which work well when run offline. First, there are no baselines (such as the method of Klindt et al.) to which FNP can be compared for accuracy and sample efficiency. Second, the method is evaluated on very specific simulated data: the classic simple/complex cell model of primary visual cortex. There is no evidence that this method would work equally well when applied, e.g., to responses in higher visual cortex -- which are notoriously hard to parametrize in a simple feature space, and on which existing methods excel. Of course, these responses are hard if not impossible to simulate in the way V1 responses are simulated, so a more challenging test would either require (a) comparing to neural data or (b) simulating data via a highly predictive DNN mapped to neural responses. I am also not sure that the proposed FNP method is actually more efficient than optimization-based estimates of neural tuning. The authors could bolster this claim with a back-of-the-envelope calculation, estimating how long it would take in a typical recording session to reach the required number of observations and how long it would take to train and run their method, compared to fitting linear models to pretrained deep NN layers. ### EDIT ### The authors have provided solid responses to both of these weaknesses. The comparison to Klindt2017 is appreciated, though I am curious to know if this "PNO" curve includes the (time-consuming) hyperparameter optimization that the authors hope to avoid with FNP. It would be good to explicitly say in the paper that FNP's main advantage is speed, but also may outperform even more time-consuming methods when there are fewer observations. The comparison of training/inference times is very helpful. I'd encourage the authors to put this table in the main text. Though 6 days of training on dual V100 is quite a lot of time/GPU memory! Will neuroscientists with more typical resources be able to use this method?

Correctness: Yes

Clarity: Yes, the paper is clearly written.

Relation to Prior Work: It's clear how the current method differs from existing methods at a technical level, but FNP really should be compared to at least one other method as a baseline -- on accuracy, efficiency, or ideally both.

Reproducibility: Yes

Additional Feedback: This work would be more compelling if the authors described a few ideal use cases for the intended method -- they clearly have interactive neural recording sessions in mind, in which the data are analyzed online and used to direct the experiment in some new direction (cf. Bashivan, Kar, & DiCarlo 2019.) But what exactly would they propose, and how is it limited by the existing set of methods?

[Author Response · NeurIPS 2020]

We thank all three reviewers for their valuable comments and positive feedback. All reviewers agree that our approach addresses an important problem: real-time inference of complex predictive neural models for online closed-loop experiments. The reviewers find that our paper "improve[s] the ability to estimate stimulus-response models for neurons in the brain" (R1), that it "is a very

| Network | Time |
|---------|------|
| FNP Optimization (once) | 6 days on dual V100 |
| FNP Inference for 1000 trials | **250 ms** (1080Ti) |
| PNO Readout only | $\sim$ 20 s (1080) |
| PNO CNN+Readout | $\sim$ 5 min (1080) |
| PNO + Hyperparameters | $\sim$ 12 h (1080) |

interesting paper and set of experiments" and its "use-case [...] compelling" (R2), and that our "approach is well motivated and would address an important problem" (R4). Their main concerns are: (1) our experiments are only on synthetic data (R1, R4), (2) our model is not compared to baseline models (R4), and (3) we need to present results about training and prediction time (R2, R4). We are confident that we can address all concerns and that doing so improves our results. We fit our Factorized Neural Processes (FNP) model to real neural responses from mouse V1. We find comparable predictive accuracy to state-of-the-art models and critically—and by design—predicting the response of unseen neurons is two-five orders of magnitude faster than using an optimization-based methodology. We will include these results in the paper and improve the clarity of the presentation (R1, R2) with additional technical details in supplemental materials. Because of the space limit, we can't respond to all detailed concerns, but we will fix them.

**General motivation**    Our goal is to rapidly infer a predictive model of newly recorded neurons with minimal latency for online, closed-loop experiments. Our envisioned use case (R4) is active learning of tuning properties where stimuli are selected based on current estimates (and uncertainties, R2) of the tuning function to better constrain the model and learn it more efficiently. This is not feasible online with current models, even if just parts of the model are retrained (see (3)). Prior experiments with predictive models fit them to newly acquired neurons overnight and tested them the next day (e.g. Bashivan2019 and Walker2019).

(1) **Demonstration on real data** (R1, R4)    We trained an FNP on 57,533 mouse V1 neurons responding to static scenes collected across 19 different scans. We tested the $K$-shot predictive accuracy on 1000 randomly selected neurons from a hold-out scan (i.e. never seen during training) with $K$ up to 1000 natural images to infer the tuning properties and predict responses to stimuli (a rapid network inference with no cell-specific optimization). In agreement with synthetic data, the predictive accuracy improves rapidly with the first several hundred trials and continues to improve with additional observations (see figure) establishing the utility of our method on real neural responses. The FNP also generates realistic receptive field estimates.

(2) **Comparison to baseline model** (R4)    We now compare the performance of FNP (ours) to a SOTA model in the style of Klindt2017, adapted for mouse V1, which we reference as Per Neuron Optimization (PNO). We trained it on the responses of a single scan with 4,335 neurons to 5k natural scenes. Subsequently, we froze the core CNN and fit a readout (linear+nonlinearity) to 1,000 new neurons on up to 1,000 images and measure the prediction accuracy. Excitingly, FNP generalizes well to new neurons and with 1k images is almost as accurate as PNO (which is optimized for those individual cells), and even *outperforms* it for smaller numbers of observations (see figure). Importantly, this is achieved with massive improvements in run-time (see (3)).

(3) **Long training time** (R2, R4)    We feel that there might have been a misunderstanding and we will make this more clear in the paper: FNP needs to be trained *only once* using all previously recorded data (see (1)). During experiments, a predictor is obtained for newly acquired neurons with a nearly instant, single pass through the FNP. This takes only 250ms (for 1k responses) compared to optimization which ranges from $\sim$20s (if only fitting the readout) to $\sim$12h (to optimize hyperparameters as we currently do in experiments). We summarize training times in the above table. Thus getting a predictor with an FNP is two-five orders of magnitude faster, enabling real-time predictions. This is not possible with existing methods (R4). Thus our approach allows inferring updated neural response properties within the time of a single stimulus presentation.

**Other** (R2)    $q(z|s)$ is a variational approximation to $p(z|s)$; **Two-stage method of ref[6]** is also optimization based; **Eqn 1** integration is approximated by a single sample using the reparameterization trick (c.f. Kingma 2015), MLE is not easy in deep learning; **sub-pixel interpolation** makes the loss differentiable for optimization (c.f. Spatial Transformer Networks); **X-axis** corresponds to real trials (as above). **Section 3.3** $\mathbb{R}^{H \times W \times C}$ is an image with $C$ channels after passing through the convolution and $H \times W$ refers to the spatial dimensions. It is then combined (non-linearly) with the responses to predict the location. Response units can be fluorescence (as above) or spike counts (see simulations) **Uncertainty** is currently only used as a tool for training, we imagine using entropy to drive stimulus selection; **Cell types:** good point, we did not want to emphasize categories, continuum is also possible, will revise terminology; **Fig 2 & LL** see (1),(2) for real world example. **Complex cell RFs:** Correct! it looks opposite at times due to the random phase shifts; **Code** was withheld for anonymity, will be released; **Broader impact** will be updated, good suggestions.

[Meta-Review · NeurIPS 2020]

The authors develop factorized neural processes to quickly adapt to novel neural spiking data in the few-shot regime. Reviewers agreed that this approach was novel, compelling and the application and extension of neural processes interesting. Some reviewers found that the paper could be clearer and one reviewer questioned whether the approach would generalize beyond the typical responses seen in the V1. In discussion, one reviewer raised possible positive ethical implications of this work - by requiring less neural spiking data it could mitigate suffering among the trial subjects. All-in-all the reviewers found that the paper would be interesting to the community and recommended acceptance.